# CogVLM: Visual Expert for Large Language Models

## Abstract

We introduce CogVLM, a powerful open-source visual language foundation model. Different from the popular *shallow alignment* method which maps image features into the input space of language model, CogVLM bridges the gap between the frozen pretrained language model and image encoder by a trainable visual expert module in the attention and FFN layers. As a result, CogVLM enables deep fusion of vision language features without sacrificing any performance on NLP tasks. CogVLM-17B achieves state-of-the-art performance on 10 classic cross-modal benchmarks, including NoCaps, Flicker30k captioning, RefCOCO, RefCOCO+, RefCOCOg, Visual7W, GQA, ScienceQA, VizWiz VQA and TDIUC, and ranks the 2nd on VQAv2, OKVQA, TextVQA, COCO captioning, etc., surpassing or matching PaLI-X 55B. Codes and checkpoints are available at Github.

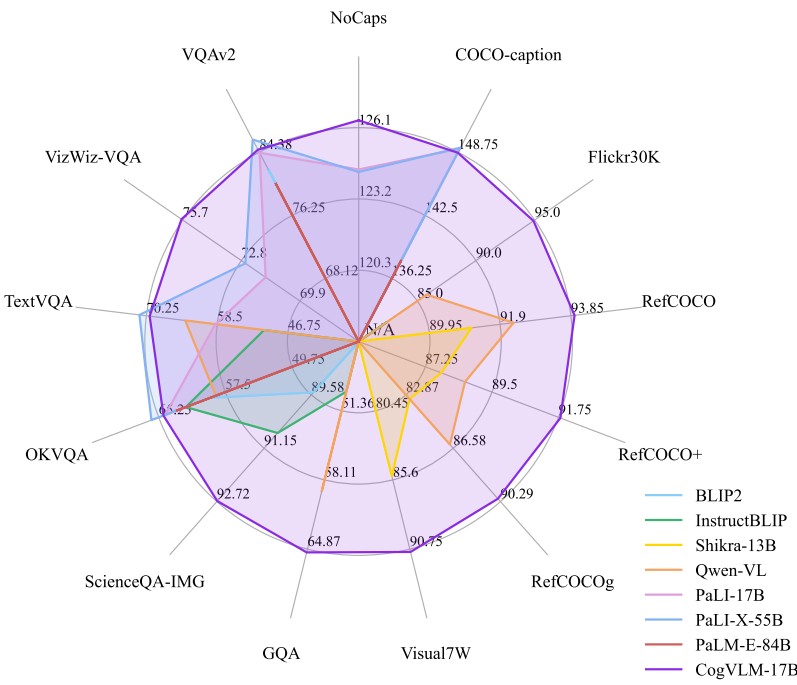

**Figure 1:** The performance of CogVLM on a broad range of multi-modal tasks in comparison with existing approaches.

# 1 Introduction

Visual language models (VLMs) are versatile and powerful. Many vision and cross-modality tasks can be formulated as next token prediction, e.g., image captioning (Agrawal et al., 2019), visual question answering (Antol et al., 2015), visual grounding (Yu et al., 2016) and even segmentation (Chen et al., 2022a). Useful abilities like in-context learning (Tsimpoukelli et al., 2021) also

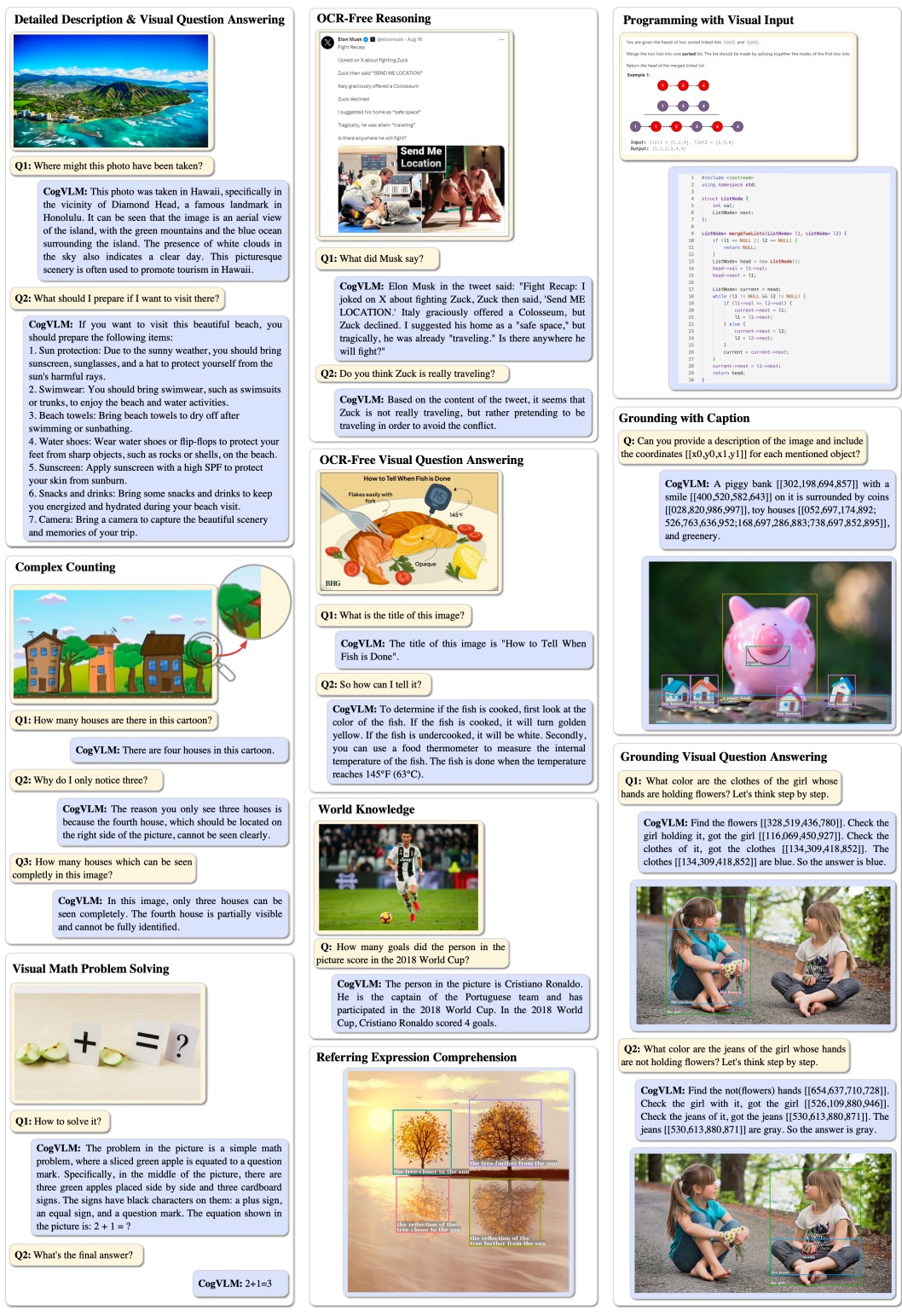

**Figure 2:** Samples generated by CogVLM.

emerge along with the improvement of downstream tasks when scaling up VLMs. However, to train a large language model is already non-trivial, and it is more challenging to train a VLM from scratch with the same NLP performance as well-trained pure language models like LLaMA2 (Touvron et al., 2023). Therefore, it is natural to investigate how to train a VLM from an off-the-shelf pretrained language model.

The popular *shallow alignment* methods represented by BLIP-2 (Li et al., 2023) connect a frozen pretrained vision encoder and language model via a trainable Q-Former or a linear layer, mapping the image features into the input embedding space of the language model. This method converges fast, but the performance (BLIP-2 NoCaps CIDEr 121.6) is not as good as jointly training the vision and language modules, e.g., PaLI-X (NoCaps CIDEr 126.3). As for chat-style VLM trained by shallow alignment methods, e.g., MiniGPT-4 (Zhu et al., 2023), LLAVA (Liu et al., 2023b), and VisualGLM, the weak visual understanding ability manifests as *hallucination*. So, is it possible to retain the NLP capabilities of the large language model while adding top-notch visual understanding abilities to it?

CogVLM gives a "*yes*" answer. In our opinion, the root cause of the inferior performance of shallow alignment methods lies in the lack of *deep fusion* between vision and language information. This inspiration arises from the comparison between p-tuning (Liu et al., 2023e) and LoRA (Hu et al., 2021) in efficient finetuning, where p-tuning learns a task prefix embedding in the input while LoRA adapts the model weights in each layer via a low-rank matrix. As a result, LoRA performs better and more stable. A similar phenomenon might also exist in VLM, because in the shallow alignment methods, the image features act like the prefix embedding in p-tuning. More detailed reasons for the performance degradation of p-tuning and shallow alignment include:

1. The frozen weights in the language model are trained for text tokens. Visual features do not have a perfect counterpart in the input text space. Therefore, after multi-layer transformations, the visual features might no longer match the input distribution of the weights in the deep layers.

2. During pretraining, the prior of the image captioning task, for example, the writing style and caption length, can only be encoded into the visual features in the shallow alignment methods. It weakens the consistency between visual features and content.

A possible solution is to adapt the language model to the image-text joint training, which is adopted by PaLI (Chen et al., 2022b) and Qwen-VL (Bai et al., 2023a). However, in this way, the NLP ability is avoidably impaired, which might affect text-centered tasks, such as image-based poetry creation or introducing the background story of images. According to PaLM-E (Driess et al., 2023), making the language model trainable during VLM pretraining will lead to catastrophic forgetting, and drop 87.3% NLG performance for 8B language model.

CogVLM instead adds a trainable *visual expert* to the language model. In each layer, the image features in the sequence use a new QKV matrix and MLP layer with the text features. Visual expert doubles the number of parameters while keeping the FLOPs the same. Since all the parameters in the original language model are fixed, the behaviors are the same as in the original language model if the input sequence contains no image.

**Our CogVLM-17B trained from Vicuna-7B achieves state-of-the-art or the second-best performance on 14 classic cross-modal benchmarks**, including 1) image captioning datasets: No-Caps, Flicker30k, COCO, 2) VQA datasets: VQAv2, OKVQA, GQA, TextVQA, VizWiz, 3) visual grounding datasets: RefCOCO, RefCOCO+, RefCOCOg, Visual7W, 4) multiple choice datasets: TDIUC, ScienceQA.

Since **most previous famous VLMs are close-source**, including Flamingo (Alayrac et al., 2022), SimVLM (Wang et al., 2021), Coca (Yu et al., 2022), BEIT-3(1.9B) (Wang et al., 2022c), GIT2 (Wang et al., 2022a), PaLI (Chen et al., 2022b), PaLI-X (Chen et al., 2023b), **we anticipate that the open-sourcing of CogVLM will greatly help the research and industrial application of visual understanding.**

## 2 METHOD

### 2.1 ARCHITECTURE

CogVLM model comprises four fundamental components: a vision transformer (ViT) encoder, an MLP adapter, a pretrained large language model (GPT), and a visual expert module. Figure 3 shows an overview of the CogVLM architecture. The components' design and implementation details are provided below:

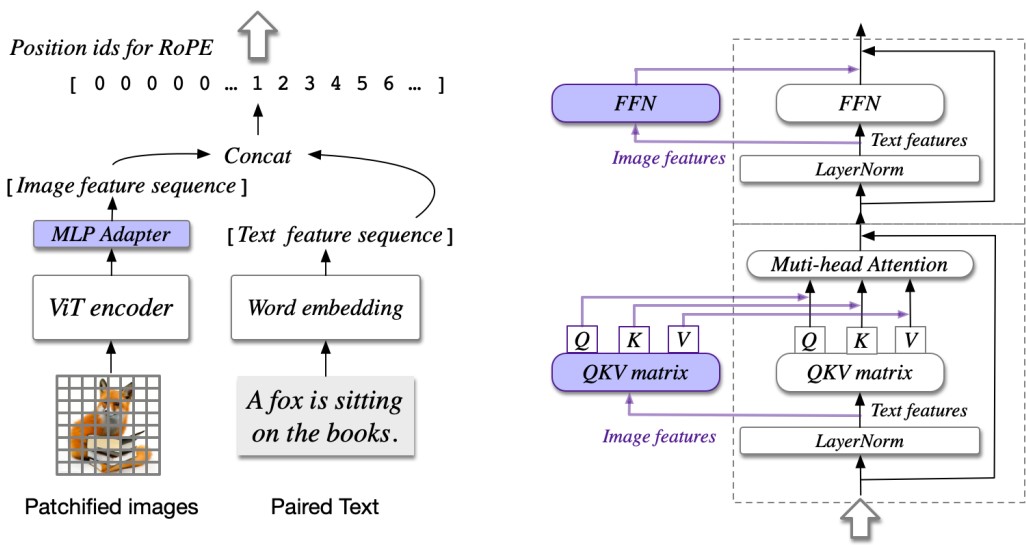

(a) The input of visual language model   (b) The visual expert built on the language model

**Figure 3:** The architecture of CogVLM. (a) The illustration about the input, where an image is processed by a pretrained ViT and mapped into the same space as the text features. (b) The Transformer block in the language model. The image features have a different QKV matrix and FFN. Only the purple parts are trainable.

**ViT encoder**. We utilize pretrained EVA2-CLIP-E (Sun et al., 2023) in CogVLM-17B. The final layer of ViT encoder is removed because it specializes in aggregating the [CLS] features for contrastive learning.

**MLP adapter**. The MLP adapter is a two-layer MLP (SwiGLU (Shazeer, 2020)) to map the output of ViT into the same space as the text features from word embedding. All image features share the same position id in the language model.

**Pretrained large language model**. CogVLM's model design is compatible with any off-the-shelf GPT-style pretrained large language model. Specifically, CogVLM-17B adopts Vicuna1.5-7B (Chiang et al., 2023) for further training. A causal mask is applied to all the attention operations, including the attention between image features.

**Visual expert module**. We add a visual expert module to each layer to enable deep visual-language feature alignment. Specifically, the visual expert module in each layer consists of a QKV matrix and an MLP in each layer. The shapes of the QKV matrix and MLP are identical to those in the pretrained language model and initialized from them. The motivation is that each attention head in the language model captures a certain aspect of semantic information, while a *trainable* visual expert can transform the image features to align with the different heads, therefore enabling deep fusion.

Formally, suppose that the input hidden states of an attention layer are $X \in \mathbb{R}^{B \times H \times (L_I + L_T) \times D}$, where $B$ is the batch size, $L_I$ and $L_T$ are the lengths of image and text sequences, $H$ is the number of attention heads, and $D$ is the hidden size. In the attention with visual expert, $X$ is first split as image hidden states $X_I$ and text hidden states $X_T$, and the attention is computed as:

$$\text{Attention}(X, W_I, W_T) = \text{softmax}(\frac{\text{Tril}(QK^T)}{\sqrt{D}})V, \tag{1}$$

$$Q = \text{concat}(X_I W_I^Q, X_T W_T^Q), K = \text{concat}(X_I W_I^K, X_T W_T^K), V = \text{concat}(X_I W_I^V, X_T W_T^V), \tag{2}$$

where $W_I, W_T$ are the QKV matrices of the visual expert and original language model, and $\text{Tril}(\cdot)$ means lower-triangular mask. The visual expert in FFN layers performs similarly,

$$\text{FFN}(X) = \text{concat}(\text{FFN}_I(X_I), \text{FFN}_T(X_T)), \tag{3}$$

where $\text{FFN}_I$ and $\text{FFN}_T$ are the FFN of the visual expert and original language model.

## 2.2 PRETRAINING

**Data.** The image-text pairs for pretraining are all publicly available, including LAION-2B and COYO-700M. After removing the broken URLs, NSFW images, images with noisy captions, images with political bias and images with an aspect ratio $> 6$ or $< 1/6$, about 1.5B images are left for pretraining.

We also crafted a visual grounding dataset of 40M images. Each noun in the image caption is associated with bounding boxes to indicate the positions in the image. The construction process basically follows Peng et al., which extracts nouns via spaCy (Honnibal & Johnson, 2015) and predicts the bounding boxes using GLIPv2 (Zhang et al., 2022). The image-text pairs are sampled from LAION-115M, a subset of LAION-400M filtered by Li et al. (2023). We filter and retain a subset of 40 million images to ensure that over 75% of images contain at least two bounding boxes.

**Training.** The first stage of pretraining is for *image captioning loss*, i.e. next token prediction in the text part. We train the CogVLM-17B model on the 1.5B image-text pairs introduced above for 120,000 iterations with a batch size of 8,192. The second stage of pretraining is a mixture of image captioning and Referring Expression Comprehension (REC). REC is a task to predict the bounding box in the image given the text description of an object, which is trained in the form of VQA, i.e., "Question: Where is the *object*?" and "Answer: $[[x_0, y_0, x_1, y_1]]$". Both $x$ and $y$ coordinates range from 000 to 999, meaning the normalized position in the image. We only consider the loss of the next token prediction in the "Answer" part. We pretrain the second stage for 60,000 iterations with a batch size of 1,024 on the text-image pairs and visual grounding datasets introduced above. During the final 30,000 iterations, we change the input resolution from $224 \times 224$ to $490 \times 490$. The total number of trainable parameters is 6.5B and the pretraining consumes about 4,096 A100$\times$days.

## 2.3 ALIGNMENT

We further finetune CogVLM on a broad range of tasks, so as to align CogVLM with free-form instructions of any topic. We name the finetuned model *CogVLM-Chat*. As the examples in Figure 2 and Appendix show, CogVLM-Chat can successfully align with diverse instructions, thus enabling flexible interaction with humans.

**Data.** The high-quality data for supervised finetuning (SFT) is collected from LLaVA-Instruct (Liu et al., 2023b), LRV-Instruction (Liu et al., 2023a), LLaVAR Zhang et al. (2023) and an in-house dataset, with a total of about 500,000 VQA pairs. The quality of SFT data is of vital importance, but the LLaVA-Instruct is generated by a pipeline involving language-only GPT-4 so that errors are inevitable. Particularly, we corrected the errors in the LLaVA-Instruct dataset via manual inspection and annotation.

**SFT.** For supervised finetuning, we train 8,000 iterations with a batch size of 640, a learning rate of $10^{-5}$ and 50 warm-up iterations.

In order to prevent overfitting the text answer of the dataset, we leverage a smaller learning rate (10% the learning rate of the other parameters) to update the pretrained language model. All the parameters except ViT encoder are trainable during SFT.

## 3 EXPERIMENTS

To rigorously validate the superior performance and robust generalization of our base model, we conduct quantitative evaluations on an array of multi-modal benchmarks. These benchmarks can be categorized into three broad areas covering a comprehensive range of measurement[1]:

- **Image Captioning**. The main purpose of these tasks is to generate textual captions summarizing the major content of a given image. We utilize prominent datasets including No-Caps (Agrawal et al., 2019), COCO (Lin et al., 2014), Flickr30K (Plummer et al., 2015), and TextCaps (Sidorov et al., 2020) for evaluation.

- **Visual Question Answering**. The VQA tasks require models to answer questions that may focus on distinct visual contents based on the given image. Our assessment covers

---

[1]Detailed summary of all benchmarks and corresponding metrics are available at Appendix A.2.

diverse datasets, including VQAv2 (Antol et al., 2015), OKVQA (Marino et al., 2019), TextVQA (Singh et al., 2019), VizWiz-VQA (Gurari et al., 2018), OCRVQA (Mishra et al., 2019), ScienceQA (Lu et al., 2022b), and TDIUC (Shrestha et al., 2019).

- **Visual Grounding**. Visual grounding involves a set of tasks that establish referential links between textual mentions in a sentence and specific regions in an image. We evaluate our model on the typical datasets, including Visual7w (Zhu et al., 2016), RefCOCO (Liu et al., 2017), RefCOCO+, and RefCOCOg to ensure completeness.

## 3.1 IMAGE CAPTIONING

We evaluate the image captioning capability of our pretrained base model on the aforementioned four benchmarks. In a zero-shot evaluation on the Nocaps and Flickr datasets, we assess the precision of our model in describing long-tail visual concepts. Additionally, we present results from finetuning on the COCO and TextCaps datasets.

The detailed performance is shown in Table 1. Overall, our model achieves the SOTA or compatible performance across the board. Specifically, on the NoCaps benchmark, our base model outperforms the previous best method, GIT2, across four splits with a maximum of 5.7 points in the out-domain set while only consuming 10% of the pretraining data (1.5B vs 12.9B). On the Flickr benchmark, our model achieves a SOTA score of 94.9 surpassing the concurrently released Qwen-VL model by 9.1 points. These results demonstrate a remarkable capability and robustness of our pretrained model on the image captioning task. We also evaluate on the COCO (Lin et al., 2014) and TextCaps, where the latter is specifically designed to integrate the textual information of the given image into captions. Though training without the dedicated OCR data, encouragingly, our base model reveals a significant text-reading ability and obtains a competitive performance with PaLI-X-55B, and outperforms the previous best model of the same scale, PaLI-17B, by 9.1 points score.

**Table 1:** Performance on Image Captioning benchmarks, where all tasks use CIDEr as the evaluation metric. OOD refers to out-of-domain test set. Karp. refers to the Karpathy test split.

| Method | Train Data | NoCaps val | | NoCaps test | | Flickr | COCO | TextCaps |
| --- | --- | --- | --- | --- | --- | --- | --- | --- |
| | | OOD | overall | OOD | overall | Karp. | Karp. | test |
| Human | - | 95.7 | 87.1 | 91.6 | 85.3 | - | - | 125.1 |
| VinVL (Zhang et al., 2021) | 8.9M | 83.8 | 94.3 | 78.0 | 92.5 | - | 130.8 | - |
| SimVLM (Wang et al., 2021) | 1.8B | 115.2 | 112.2 | 109.5 | 110.3 | - | 143.3 | - |
| CoCa (Yu et al., 2022) | 4.8B | - | 122.4 | - | 120.6 | - | 143.6 | - |
| LEMON (Hu et al., 2022) | 2B | 120.2 | 117.3 | 110.1 | 114.3 | - | 139.1 | - |
| Flamingo (Alayrac et al., 2022) | 2.3B | - | - | - | - | 67.2 | 138.1 | - |
| Prismer (Liu et al., 2023c) | 12.7M | 113.5 | 112.9 | - | 110.8 | - | 136.5 | - |
| BLIP-2 (Li et al., 2023) | 129M | 124.8 | 121.6 | - | - | - | 144.5 | - |
| InstructBLIP (Dai et al., 2023) | 129M | - | 123.1 | - | - | 82.4 | - | - |
| UniversalCap (Cornia et al., 2021) | 35M | 123.4 | 122.1 | 114.3 | 119.3 | - | 143.4 | - |
| GIT (Wang et al., 2022a) | 0.8B | 127.1 | 125.5 | 122.0 | 123.4 | 49.6 | 144.8 | 138.2 |
| GIT2 (Wang et al., 2022a) | 12.9B | 130.6 | 126.9 | 122.3 | 124.8 | 50.7 | 145.0 | 145.0 |
| Qwen-VL (Bai et al., 2023a) | 1.4B | - | 121.4 | - | - | 85.8 | - | - |
| PaLI-17B (Chen et al., 2022b) | 1.6B | - | 127.0 | - | 124.4 | - | 149.1 | 135.4 |
| PaLI-X-55B (Chen et al., 2023b) | - | - | 126.3 | - | 124.3 | - | **149.2** | **147.0** |
| CogVLM (ours) | 1.5B | **132.6** | **128.3** | **128.0** | **126.4** | **94.9** | 148.7 | 144.9 |

## 3.2 VISUAL QUESTION ANSWERING

Visual Question Answering is a task of validating general multi-modal capabilities of models, which requires a mastery of skills including vision-language understanding and commonsense reasoning. We evaluate our model on 7 VQA benchmarks: VQAv2, OKVQA, GQA, VizWiz-QA, OCRVQA, TextVQA, ScienceQA, covering a wide range of visual scenes. We train our base model on the

training sets and evaluate it on the publicly available val/test sets for all benchmarks, where both procedures adopt the open-vocabulary generation settings without OCR pipeline input.

**Table 2:** Performance on Visual Question Answering benchmarks, where the results labeled with * refers to the few-shot or zero-shot setting.

| Method | VQAv2 | | OKVQA | GQA | VizWizQA | | OCRVQA | TextVQA | SciQA |
|---|---|---|---|---|---|---|---|---|---|
| | test-dev | test-std | val | test-balanced | test-dev | test-std | test | test | IMG |
| *Closed-ended classification models* | | | | | | | | | |
| SimVLM (Wang et al., 2021) | 80.0 | 80.3 | - | - | - | - | - | - | - |
| CoCa (Yu et al., 2022) | 82.3 | 82.3 | - | - | - | - | - | - | - |
| OFA (Wang et al., 2022b) | 82.0 | 82.0 | - | - | - | - | - | - | - |
| BEiT-3 Wang et al. (2022c) | 84.2 | 84.0 | - | - | - | - | - | - | - |
| *Open-ended generation models* | | | | | | | | | |
| GIT (Wang et al., 2022a) | 78.6 | 78.8 | - | - | 68.0 | 67.5 | 68.1 | 59.8 | - |
| GIT2 (Wang et al., 2022a) | 81.7 | 81.9 | - | - | 71.0 | 70.1 | 70.3 | 67.3 | - |
| Flamingo-80B (Alayrac et al., 2022) | 82.0 | 82.1 | 57.8* | - | 65.7 | 65.4 | - | 54.1 | - |
| BLIP-2 (Li et al., 2023) | 82.2 | 82.3 | 59.3 | 44.7* | - | - | 72.7 | - | 89.5 |
| InstructBLIP (Dai et al., 2023) | - | - | 62.1 | 49.5* | 34.5* | - | 73.3 | 50.7* | 90.7 |
| PaLI-17B Chen et al. (2022b) | 84.3 | 84.3 | 64.5 | - | 71.6 | 70.7 | - | 58.8 | - |
| PaLI-X-55B (Chen et al., 2023b) | **86.0** | **86.1** | **66.1** | - | 72.6 | 70.9 | **75.0** | **71.4** | - |
| PaLM-E-84B (Driess et al., 2023) | 80.5 | - | 63.3 | - | - | - | - | - | - |
| CogVLM (ours) | 84.7 | 84.7 | 64.7 | **65.2** | 76.4 | 75.8 | 74.5 | 69.7 | **92.7** |

As shown in Table 2, our model achieves state-of-the-art performance on 6 of 7 benchmarks compared with models of similar scales, such as PALI-17B and Qwen-VL. Our model even surpasses models of much larger scale on multiple benchmarks, such as PaLI-X-55B on VizWiz-QA (test-std +5.1, test-dev +3.8), PALM-E-84B on VQAv2 (test-dev +4.2) and OKVQA(+1.4), Flamingo-80B on VQAv2 (test-dev +2.7, test-std +2.6), VizWiz-QA (test-dev +10.7, test-std +10.4) and TextVQA (+15.6). Our model also achieves the optimal scores of 92.71 on the multi-modal split (*i.e.,* IMG) of ScienceQA (Lu et al., 2022b), achieving a new SOTA. These results suggest that our base model can serve as a strong multi-modal backbone capable of solving various visual question answering tasks.

**Generalist performance.** In order to fairly compare with Unified-IO (Lu et al., 2022a), Qwen-VL (Bai et al., 2023a), mPLUG-DocOwl (Ye et al., 2023) and other models trained in a generalist paradigm across multi-modal tasks, we further trained a unified model using data composed of dozens of multi-modal datasets and utilized a consistent checkpoint for evaluation. The datasets encompass 14 QA datasets such as VQAv2, OKVQA, and extending to TextVQA, as well as caption datasets including COCO caption, TextCaps, and those used during the pre-training phase. Experimental results show that multitask learning does not significantly reduce the model's performance on individual tasks, and CogVLM remains leading in performance across all tasks.

**Table 3:** Generalist performance on Image Captioning and VQA benchmarks.

| Method | COCO | TextCaps | NoCaps | Flickr | VQAv2 | OKVQA | TextVQA | OCRVQA |
|---|---|---|---|---|---|---|---|---|
| | Karp.-test | val | val | Karp.-test | test-dev | val | val | test |
| Qwen-VL (Bai et al., 2023a) | - | - | 121.4 | 85.8 | 79.5 | 58.6 | 63.8 | **75.7** |
| mPLUG-DocOwl (Ye et al., 2023) | - | 111.9 | - | - | - | - | 52.6 | - |
| Unified-IO (Lu et al., 2022a) | 122.3 | - | 100.0 | - | 77.9 | 54.0 | - | - |
| CogVLM (single task) | 148.7 | 149.8 | 128.3 | 94.9 | 84.7 | 64.7 | 69.3 | 74.5 |
| CogVLM (generalist) | **147.0**(-1.7) | **151.3**(+1.5) | **126.2**(-2.1) | **92.7**(-2.2) | **83.4**(-1.3) | **58.9**(-5.8) | **68.1**(-1.2) | 74.1(-0.4) |

### 3.3 Visual Grounding

In order to endow our model with consistent, interactive visual grounding capabilities, we collect a high-quality dataset covering 4 types of grounding data: (1) **Grounded Captioning (GC)** - image captioning datasets where each noun phrase within the caption is followed by the corresponding referential bounding boxes; (2) **Referring Expression Generation (REG)** - image-oriented datasets that each bounding box in the image is annotated with a descriptive textual expression that accurately characterizes and refers to the content within the specific region; (3) **Referring Expression Comprehension (REC)** - text-oriented datasets that each textual description is annotated with multiple referential links associating the phrases with corresponding boxes; (4) **Grounded Visual Question Answering (GroundedVQA)** - VQA-style datasets where the questions may contain region references in a given image. The sources of grounding data are all publicly available, including Flickr30K Entities (Plummer et al., 2015), RefCOCO (Kazemzadeh et al., 2014; Mao et al., 2016; Yu et al., 2016), Visual7W (Zhu et al., 2016), VisualGenome (Krishna et al., 2017) and Grounded CoT-VQA (Chen et al., 2023a). $[box]$ in this section is in the format of $[[x_0, y_0, x_1, y_1]]$.

After the second pretraining stage using our 40M visual grounding dataset, we continue to train our model on this high-quality dataset, resulting in a generalist grounding-enhanced model, CogVLM-grounding. It is noteworthy that the curated datasets exhibit a versatility of visual grounding capabilities, and many datasets can be adapted and repurposed across different tasks. For instance, grounded captioning datasets can be reformulated to suit REG and REC tasks. Taking the example of *"A man $[box_1]$ and a woman $[box_2]$ are walking together."*, this can be reframed into question answering pairs like *("Describe this region $[box_2]$.", "A woman.")* and *("Where is the man?", "$[box_1]$")*. Similarly, REC datasets can be translated into REG tasks by switching the input and output, and vice versa. However, certain conversions might lead to ambiguities. For example, when presented with the isolated query "Where is another man?" from the caption "A man $[box_1]$ is running, while another man $[box_2]$ is looking.", the distinction between $[box_1]$ and $[box_2]$ becomes unclear, potentially leading to errors.

Table 4 shows the result on the standard visual grounding benchmarks. We find that our generalist model achieves state-of-the-art performance across the board, with a significant advantage over the previous or concurrent models. Moreover, we also evaluate the specialist performance of our model finetuned on each individual training set of benchmarks for fair comparison with the best models dedicated on each task. As shown in the bottom part of Table 4, our model achieves the SOTA performance over 5 of 9 splits, and the compatible result on the other subsets. These results suggest a remarkable visual grounding capability of our model incorporating our training paradigm.

**Table 4:** Results on Referring Expression Comprehension and Grounded Visual Question Answering.

| Type | Model | RefCOCO | | | RefCOCO+ | | | RefCOCOg | | Visual7W |
|---|---|---|---|---|---|---|---|---|---|---|
| | | val | test-A | test-B | val | test-A | test-B | val | test | test |
| *Generalist* | OFA-L* (Wang et al., 2022b) | 79.96 | 83.67 | 76.39 | 68.29 | 76.00 | 61.75 | 67.57 | 67.58 | - |
| | VisionLLM-H (Wang et al., 2023b) | - | 86.70 | - | - | - | - | - | - | - |
| | Shikra-7B (Chen et al., 2023a) | 87.01 | 90.61 | 80.24 | 81.60 | 87.36 | 72.12 | 82.27 | 82.19 | - |
| | Shikra-13B | 87.83 | 91.11 | 81.81 | 82.89 | 87.79 | 74.41 | 82.64 | 83.16 | 85.33 |
| | Qwen-VL (Bai et al., 2023a) | 89.36 | 92.26 | 85.34 | 83.12 | 88.25 | 77.21 | 85.58 | 85.48 | - |
| | **CogVLM** | **92.51** | **93.95** | **88.73** | **87.52** | **91.81** | **81.43** | **89.46** | **90.09** | **90.96** |
| *Specialist* | G-DINO-L Liu et al. (2023d) | 90.56 | 93.19 | 88.24 | 82.75 | 88.95 | 75.92 | 86.13 | 87.02 | - |
| | UNINEXT-H (Lin et al., 2023) | 92.64 | **94.33** | **91.46** | 85.24 | 89.63 | 79.79 | 88.73 | 89.37 | - |
| | ONE-PEACE (Wang et al., 2023a) | 92.58 | 94.18 | 89.26 | **88.77** | 92.21 | **83.23** | 89.22 | 89.27 | - |
| | **CogVLM (single task)** | **93.40** | 94.06 | 90.28 | 87.76 | **93.02** | 81.81 | **90.07** | **90.53** | **91.17** |

### 3.4 Instruction Following in Real-world User Behavior

To evaluate the CogVLM-Chat model's capacity under real-world user behavior, we further employ TouchStone (Bai et al., 2023b), an extensive benchmark for multimodal language models. Table 5 shows the GPT-4 (OpenAI, 2023) similarity scores of the generated and standard answer, suggesting CogVLM-Chat significantly outperforms all the other publicly available VLMs.

**Table 5:** Evaluation results on TouchStone in English.

| Models | MiniGPT4 | InstructBLIP | LLaMA-AdapterV2 | LLaVA | mPLUG-Owl | Qwen-VL-Chat | **CogVLM-Chat** |
|---|---|---|---|---|---|---|---|
| **Score** | 531.7 | 552.4 | 590.1 | 602.7 | 605.4 | 645.4 | **662.6** |

**Table 6:** Ablation studies for various components and training settings.

| Ablated Aspects | Original (CogVLM) | Ablated Setting | Trainable params | COCO CIDEr↑ | NoCaps CIDEr↑ | OKVQA top1↑ | TextVQA top1↑ | VQAv2 top1↑ |
|---|---|---|---|---|---|---|---|---|
| Tuned Parameters | *VE-full* every layer + MLP Adapter | MLP Adapter | 140M | 131.2 | 111.5 | 55.1 | 40.7 | 73.8 |
| | | LLM+MLP Adapter | 6.9B | 140.3 | 118.5 | 56.8 | 44.7 | 78.9 |
| | | *VE-full* every 4th layer | 1.7B | 138.7 | 117.4 | 58.9 | 44.1 | 77.6 |
| | | *VE-FFN* every layer | 4.4B | 140.0 | 118.7 | 58.2 | 45.1 | 78.6 |
| Init method | From LLM | Random init | 6.6B | 138.0 | 117.9 | 55.9 | 44.0 | 79.1 |
| Visual attention mask | Causal mask | Full mask | 6.6B | 141.0 | 117.2 | 57.4 | 45.1 | 79.6 |
| Image SSL loss | ✗ | ✓(clip feature) | 6.6B | 142.9 | 119.8 | 58.7 | 45.9 | 79.7 |
| EMA | ✓ | ✗ | 6.6B | 143.1 | 119.2 | 57.1 | 43.8 | 79.4 |
| *CogVLM (ours)* | — | — | 6.6B | 142.8 | 120.1 | 59.3 | 45.3 | 80.0 |

## 3.5 ABLATION STUDY

To understand the impact of various components and settings on our model's performance, we conduct an extensive ablation study for 6,000 iterations and a batch size of 8,192. Table 6 summarizes the results about the following aspects:

**Model structure and tuned parameters**. We investigate the effectiveness of tuning only the MLP Adapter layer or tuning all LLM parameters and the Adapter without adding VE, as well as modifying the VE architecture to add full VE at every 4th LLM layer or only the FFN-equipped VE at all layers. From the results we can see that only tuning the adapter layer (*e.g.,* BLIP2) may result in a shallow alignment with significantly inferior performance, and decreasing either the number of VE layers or the VE parameters at each LLM layer suffers a prominent degradation.

**Initialization Method**. We investigate the effectiveness of initializing VE weights from LLM, and the slight decrease in performance suggests a positive impact of this method.

**Visual Attention Mask**. We empirically find that using a causal mask on visual tokens will yield a better result in comparison with a full mask. We hypothesize the possible explanation for this phenomenon is that the causal mask better fits the inherent structure of LLM.

**Image SSL Loss**. We also investigated the self-supervised learning loss on image features, where each visual feature predicts the CLIP feature of the next position for visual self-supervision. Align with the observation from PaLI-X (Chen et al., 2023b), we find it brings no improvement on downstream tasks, although we indeed observed improvements in small models in our early experiments.

**EMA**. We utilize EMA (Exponential Moving Average) during pretraining, which often brings improvements across various tasks.

## 4 CONCLUSION

In this paper, we introduce CogVLM, an open visual language foundation model. CogVLM shifts the paradigm for VLM training from shallow alignment to deep fusion, achieving state-of-the-art performance on 10 classic multi-modal benchmarks.

The VLM training is still in its infancy, and there are many directions to explore, for example, better SFT alignment, RLHF and anti-hallucination. Since the previous famous VLMs are mostly closed-source, we believe CogVLM will be a solid foundation for future multi-modal research.

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

## A    APPENDIX

### A.1    DETAILS OF TRAINING SETTINGS

We report the details of parameter settings during pre-training and multitask training in Table 7 and Table 8.

Table 7: Hyperparameters for pre-training model.

| Hyperparameters | Stage 1 | Stage 2 |
|---|---|---|
| Total steps | $120,000$ | $60,000$ |
| Warmup steps | $12,000$ | $1,200$ |
| Batch size | $8,192$ | $1,024$ |
| Learning rate | $1e^{-4}$ | $1e^{-5}$ |
| Learning rate decay | Cosine | |
| Weight decay | 0.05 | |
| Dropout ratio | 0.1 | |
| Adam $\epsilon$ | $1e^{-8}$ | |
| Adam $\beta$ | (0.9, 0.95) | |
| Textual encoder | Vicuna-1.5-7B | |
| Visual encoder | EVA2-CLIP-E | |
| Patch size | 14 | |
| Input resolution | $224^2$ | $224^2 \to 490^2$ |

Table 8: Hyperparameters for multitask finetuning CogVLM.

| Hyperparameters | Multitask |
|---|---|
| Learning rate | $1e^{-5}$ |
| Total steps | 10,000 |
| Batch size | 1,024 |
| AdamW $\epsilon$ | $1e^{-8}$ |
| AdamW $\beta$ | (0.9, 0.95) |
| Weight decay | 0.1 |
| Dropout ratio | 0.1 |
| Input resolution | $490^2$ |

### A.2    DETAILS OF ASSOCIATED DATASETS

In this section, we introduce the details of datasets and their use in our evaluation process for all associated benchmarks.

#### A.2.1    IMAGE CAPTIONING

- **COCO** (Lin et al., 2014) The Captions in COCO dataset are collected using Amazon's Mechanical Turk (AMT) workers who are given instructions to control the quality. The dataset contains 330K images, where the train, validation and test sets contain 413,915 captions for 82,783 images, 202,520 captions for 40,504 images, and 379,249 captions for 40,775 images respectively.

- **NoCaps** (Agrawal et al., 2019). NoCaps is a large-scale benchmark for novel object captioning, containing nearly 400 novel object classes compared to COCO. The validation and test set comprised of 4,500 and 10,600 images, respectively, sourced from the Open

**Table 9:** Summary of the evaluation benchmarks.

| Task | Dataset | Description | Split | Metrics |
|------|---------|-------------|-------|---------|
| Image Caption | NoCaps | Captioning of natural images. | val | CIDEr (↑) |
| | Flickr | Captioning of natural images. | karpathy-test | CIDEr (↑) |
| | COCO | Captioning of natural images. | karpathy-test | CIDEr (↑) |
| | TextCaps | Captioning of natural images containing text. | test | CIDEr (↑) |
| General VQA | VQAv2 | VQA on natural images. | test-dev | VQA Score(↑) |
| | OK-VQA | VQA on natural images requiring outside knowledge. | val | VQA Score (↑) |
| | GQA | VQA on scene understanding and reasoning. | test-dev-balanced | EM (↑) |
| | VizWiz-QA | VQA on photos taken by people who are blind. | test-dev | VQA Score (↑) |
| | ScienceQA | Multi-choice VQA on a diverse set of science topics | test | Accuracy (↑) |
| | TDIUC | VQA on natural images with detailed question types. | val | VQA Score (↑) |
| Text-oriented VQA | OCR-VQA | VQA on images of book covers. | test | EM (↑) |
| | TextVQA | VQA on natural images containing text. | val | VQA Score (↑) |
| Grounding | RefCOCO | Refer grounding on natural images. | test-B | Accuracy (↑) |
| | RefCOCO+ | Refer grounding on natural images. | test-B | Accuracy (↑) |
| | RefCOCOg | Refer grounding on natural images. | test | Accuracy (↑) |
| | Visual7W | VQA with referential regions selection. | val | Accuracy (↑) |

Images (Krasin et al., 2017) and annotated with 11 human-generated captions per image, and each set is subdivided into three domains: "in", "near", and "out", with objects in the "out-domain" never appearing in the COCO dataset.

- **Flickr30K** (Plummer et al., 2015). Flickr30K is a high-quality dataset consists of 31,783 images of everyday life activities, envets and scenes (all harvested from the online website Flickr) and 158,915 captions (obtained via crodsourcing). Each image in this dataset is described independently by five annotators who are not familiar with the specific entities and circumstances depicted in them.

- **TextCaps** (Sidorov et al., 2020) Textcaps is a dataset with 145k captions for 28k images. The design purpose of the TextCaps dataset is to effectively integrate textual information with visual context into captions, requiring the model to have both excellent OCR capabilities and strong captioning abilities.

### A.2.2 GENERAL VQA

- **VQAv2** (Antol et al., 2015) VQAv2 encompasses over 200,000 images, paired with more than 1.1 million questions that have collectively garnered over 11 million answers. Questions span various types, including yes/no, counting, and open-ended queries.

- **OKVQA** (Marino et al., 2019) The OK-VQA (Outside Knowledge Visual Question Answering) dataset is specifically designed to probe visual question answering capabilities that necessitate external knowledge or common sense beyond image content. It has 14,055 open-ended questions and 5 ground truth answers per question.

- **VizWiz-VQA** (Gurari et al., 2018) The VizWiz-VQA dataset is derived from blind individuals capturing images and voicing related questions, accompanied by 10 crowdsourced responses per query. The central challenge of this dataset involves predicting the visual question's answer and determining if it's unanswerable.

- **ScienceQA** (Lu et al., 2022b) The ScienceQA dataset comprises 21,208 multimodal multiple-choice questions spanning three diverse subjects: natural science, language science, and social science. Each question is annotated with explanations linked to relevant lectures.

- **TDIUC** (Shrestha et al., 2019) The TDIUC dataset features 1.6M questions across 170K images from MS COCO and Visual Genome. Categorized into 12 distinct question types, it ranges from basic tasks like identifying objects or colors to more advanced reasoning like counting or positional discernment.

### A.2.3 TEXT-ORIENTED VQA

- **OCRVQA** (Mishra et al., 2019) OCR-VQA consists of 207,572 book cover images with over 1 million question-answer pairs.

- **TextVQA** (Singh et al., 2019) TextVQA is a dataset with 45,336 questions on 28,408 images that challenges models to detect, read, and reason about text within images to provide answers.

### A.2.4 GROUNDING

- **RefCOCO/RefCOCO+** (Liu et al., 2017) RefCOCO and RefCOCO+ evolved from the ReferItGame. Both subsets focus on images with two or more similar objects. RefCOCO, with 142,209 expressions across 19,994 images, places no linguistic constraints. Conversely, RefCOCO+ emphasizes appearance-centric descriptions, omitting locational terms, and comprises 141,564 expressions over 19,992 images.

- **RefCOCOg** Mao et al. (2016) The RefCOCOg subset was amassed through Amazon Mechanical Turk, where workers penned natural referring expressions for objects in MSCOCO images; it boasts 85,474 referring expressions spanning 26,711 images, each containing 2 to 4 objects of the same category.

- **Visual7W** (Zhu et al., 2016). The Visual7W dataset is predominantly designed for VQA tasks, with a dedicated subset crafted for grounded VQA. In this subset, models are presented with an image accompanied by a "which"-type question, such as "Which is the small computer in the corner?". Participants are then given four bounding boxes within the image, from which they must select the correct one as the answer. The grounded Visual7W part consists of 25,733 images and 188,068 questions.

- **Flickr30K-Entities** (Plummer et al., 2015). The Flickr30K Entities dataset, a precursor in the realm of grounded captioning, encompasses a collection of 31,783 images accompanied by 158k captioning annotations. Every caption in this dataset has been meticulously annotated such that each noun phrase is linked with a manually delineated referential bounding box. In total, there are 276k such annotated bounding boxes provided within this dataset.

- **VisualGenome (Krishna et al., 2017).** The VisualGenome dataset stands as a cornerstone in understanding the multifaceted relationships present within images. With a collection of over 100k images, each image is annotated in detail, capturing an average of 21 objects, 18 attributes, and 18 inter-object relationships. A unique aspect of this dataset is the alignment of objects, attributes, relationships, and region descriptions with standardized terminologies from WordNet. Specifically tailored for the REG and REC tasks, each annotated region in an image comes with a corresponding descriptive text, making it a rich resource for image understanding and semantic modeling. We use the subset with around 86k images and 3.6 million region-caption pairs for visual grounding.

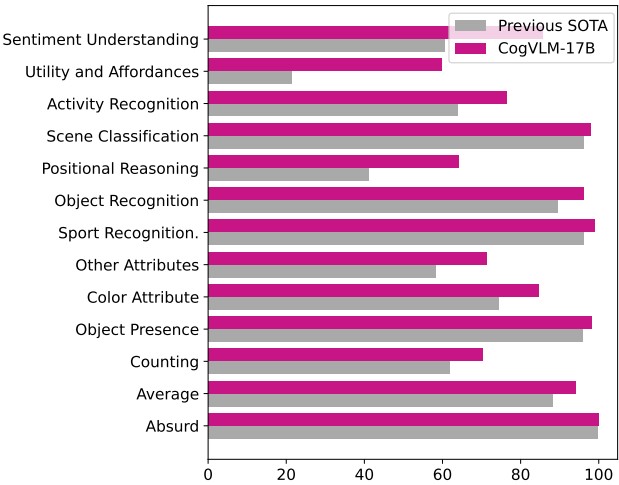

**Figure 4:** Performance on TDIUC benchmark with fine-grained questions classes.

## B  ADDITIONAL FINE-GRAINED EXPERIMENTS

To comprehensively investigate the proposed model on specific topics and question types, we further conduct extensive experiments on a representative benchmark, TDIUC (Kafle & Kanan, 2017). We use the publicly available split of val set as evaluation data, and the VQA accuracy calculated from their official scripts as the evaluation metric.

The experimental results on TDIUC compare our model against the specialist SOTA method MUREL (Cadene et al., 2019) are shown in Figure 4. From the experimental result, we can see that our model consistently outperforms the previous model on 12 specific question types, resulting in a 94.0 accuracy score compared to the previous SOTA of 88.2 on the overall dataset. These results demonstrate that our model exhibits comprehensive problem-solving skills on general VQA tasks.

## C  COMPUTATIONAL EFFICIENCY

In this section, we compare the computational efficiency of our model with other state-of-the-art models, considering both pretraining and finetuning data from datasets such as VQAv2 and TextVQA. Owing to an optimized architecture and the utilization of high-quality pretraining data, our model demonstrates a marked reduction in resource consumption during training relative to models with comparable parameter magnitudes.

**Table 10:** Comparison of different models based on their computational efficiency. We use PFLOPS*days as metrics.

| Model | Pretraining Data | Pretraining compute | VQAv2 finetuning | TextVQA finetuning |
|---|---|---|---|---|
| PaLI-3B | 1.6B | 56 | 1.1 | 0.2 |
| PaLI-17B | 1.6B | 453 | 4.5 | 0.9 |
| Flamingo-80B | 2.3B | 1381* | N/A | N/A |
| GIT2-5.1B | 12.9B | 5513* | N/A | N/A |
| CogVLM | 1.5B | 230.1 | 1.2 | 0.13 |

