# OpenReview forum: "CogVLM: Visual Expert for Large Language Models"
_ICLR.cc/2024/Conference — Submitted to ICLR 2024_

### Official Review · Reviewer_U75G · 2023-10-28

**Soundness:** 4 excellent
**Presentation:** 4 excellent
**Contribution:** 3 good
**Rating:** 6
**Confidence:** 4

**Summary:**

This paper introduces an innovative VLM framework that integrates a pre-trained image encoder and a language model through a deeper fusion and fine-tuning process. The method has demonstrated SOTA performance on multiple Vision-Language benchmarks. Furthermore, the model incorporates an alignment stage to further enhance its capabilities.

**Strengths:**

1. The proposed method represents a novel approach to multimodal techniques, distinguishing itself from previous Vision-Language Models (VLMs) like Flamingo and PaLI. The method's innovative and effective feature fusion into the language model sets it apart.

2. The proposed method has achieved SOTA performance on a range of Vision-Language benchmarks, spanning image captioning, Visual Question Answering, and visual grounding tasks.

3. The paper meticulously provides all experimental details, and the ablation study helps to validate the design components, enhancing the overall robustness of the research.

**Weaknesses:**

Comparing the proposed method to earlier approaches such as PaLI, CoCa, and Flamingo may not be entirely fair. These prior methods do not incorporate the SFT stage, making it unclear how the model performs before this crucial phase.

**Questions:**

Since the pretrained image encoder and LM and went through VLM finetuning, their original behavior may have changed. I wonder what the visual eval (linear probe, zero shot) will be for this finetuned encoder, compared to original model. How LM performance got affected?

---

> ### Author Response · Authors · 2023-11-17
> **Response**
>
> Thank you for your comprehensive review and important comments. Please allow us to address your points in detail:
>
> > 1. Comparing the proposed method to earlier approaches such as PaLI, CoCa, and Flamingo may not be entirely fair. These prior methods do not incorporate the SFT stage, making it unclear how the model performs before this crucial phase.
>
> This is a misunderstanding. We did not incorporate SFT to evaluate the benchmarks, so the evaluation is fair. See 2. of "response to common concerns" for details.
>
> > 2. Since the pretrained image encoder and LM and went through VLM finetuning, their original behavior may have changed. I wonder what the visual eval (linear probe, zero shot) will be for this finetuned encoder, compared to original model. How LM performance got affected?
>
> This is also a misunderstanding. During the pretraining, the parameters of both the ViT image encoder and the LLM were completely **frozen**, thus the ViT is not affected, because it is also frozen in SFT.  LM is trainable during SFT with a very small learning rate of 1e-6 for 8,000 iterations, which might affect the language style but not much on knowledge.
>
> We find your concerns are mainly from two important misunderstandings. Could you increase your rating if our responses clarify it?

---

> ### Author Response · Authors · 2023-11-22
> **Kindly Reminder**
>
> Dear Reviewer: Thanks again for your careful and valuable comments! Since the rebuttal discussion is due soon, we’ll be appreciated to know whether our replies have addressed your questions. If there are any further clarifications required or any other concerns, please feel free to contact us. Many thanks!

---

### Official Review · Reviewer_c9aL · 2023-10-30

**Soundness:** 3 good
**Presentation:** 4 excellent
**Contribution:** 3 good
**Rating:** 6
**Confidence:** 4

**Summary:**

This paper introduces a powerful open-source visual language foundation model, CogVLM, which adds a trainable visual expert module in the attention and FFN layers, to allow for deep fusion between visual and textual features. By pretraining on a large-scale image-text aligned data and benchmark datasets and utilizing a multi-stage training strategy, it achieves state-of-the-art performance on a variety of classic cross-modal benchmarks.

**Strengths:**

1. Strong performance on a variety of popular benchmarks, including VQA, Image Caption, Visual Grounding, Document Visual Tasks, and some GPT-4 based evaluation.
2. One pioneering work to address the shallow alignment problem for cross-modal learning by introducing visual expert module in MLLM.
3. Open-source MLLM for better promoting the cross-modal research.

**Weaknesses:**

1. The idea is not that novel, compared with BEIT-3 and VLMo, which also introduces different modality expert structures, although this work makes some changes to make it work in the era of LLM.
2. The VQA/Image Caption model, Visual Grounding model and Chat model are three different models, I am wondering how the performance can be if all these models are a unified one? Since GPT-4V may be a unified one.
3. Although I appreciate the excellent performance it achieves, the visual backbone is ViT-e, and the input resolution is 490 * 490, also the parameter size of LLM doubles, which makes the comparason a little hard.

**Questions:**

1. Do you have more experiment on the archtecture of visual expert module and more insight about which part of the layers should be shared and which module should have separate parameters?
2. For the generalist performance, is it possible that a model can achieve best performance on both real-world chat and benchmark datasets? since this paper has three separate training procedures to make it best in each individual dataset. If there exist some gaps between different kinds of datasets, how can the architecture be designed to better address this problem?
3. Have you observed some new emergent ability in this strong MLLM?

---

> ### Author Response · Authors · 2023-11-17
> **Response**
>
> Thank you for your comprehensive review and insightful comments. Please allow us to address your points in detail:
> > 1. The idea is not that novel, compared with BEIT-3 and VLMo, which also introduces different modality expert structures, although this work makes some changes to make it work in the era of LLM.
>
> Our response to this concern is in  1. of "reply to common concerns" above. The MoE is definitely not a new idea, but in the current LLM+ paradigm, it is very valuable to rediscover its effectiveness as a way to better scaling up.
>
> > 2. The VQA/Image Caption model, Visual Grounding model and Chat model are three different models, I am wondering how the performance can be if all these models are a unified one? Since GPT-4V may be a unified one.
>
> Previously, we did not unify the models for VQA, chat, and grounding tasks due to their diverse output requirements, such as single-word answers for VQA, lengthy analytical responses for chat, and coordinate-heavy answers for grounding. However, our latest experiments indicate that VQA and chat models can be effectively unified. We will soon release a unified checkpoint where simple prompt modifications like "Short answer:" and "Answer:" enable the model to perform strongly in both VQA and chat domains, as shown in our latest results in Table 1(The touchstone score increase from 663 to 740).
>
> The grounding task is slightly different; our results show it slightly reduces the model’s captioning ability. For instance, performance on the COCO dataset dropped by 2 points and on the NoCaps dataset by 1.5 points, potentially due to the model’s limited understanding of coordinates. Conversely, grounding training improved the model's performance by 0.8 point on the  VQAv2 dataset, likely due to enhanced object localization capabilities. We believe a fully unified model is achievable and are working towards this, with adaptations needed for grounding tasks, such as adding specific coordinate-representing words to the vocabulary and loosening parameters in the word embedding layer and LM head during training.
> > 3. Although I appreciate the excellent performance it achieves, the visual backbone is ViT-e, and the input resolution is 490 * 490, also the parameter size of LLM doubles, which makes the comparason a little hard.
>
> Our comparison is fair enough actually. It's common for multimodal models to upscale resolution for downstream tasks to enhance performance, similar to BLIP-2 (490 * 490) and Qwen-VL (448 * 448), with Google’s PaLI-X even using 896 * 896 resolution for image inputs.
> For ViT-E, we also experimented with the OpenCLIP-L visual encoder, comprising 300M parameters, and found its performance only slightly worse than that of ViT-E. For instance, the score on the VQAv2 test set was 83.4 with OpenCLIP-L compared to 84.7 with ViT-E, and 88.6 on the Visual-7W dataset versus 90.6 with ViT-E.
> > 4. Do you have more experiment on the archtecture of visual expert module and more insight about which part of the layers should be shared and which module should have separate parameters?
>
> In the ablation study part, we experimented this. The finding basically shows that the best way is to insert new parameters in each layer.
> > 5. For the generalist performance, is it possible that a model can achieve best performance on both real-world chat and benchmark datasets? since this paper has three separate training procedures to make it best in each individual dataset. If there exist some gaps between different kinds of datasets, how can the architecture be designed to better address this problem?
>
> Yes, we merged the training of chat and generalist on benchmarks, which is reported in the response to the second question. The results are even better on chat and basically the same for benchmarks (generalist).
>
> > 6. Have you observed some new emergent ability in this strong MLLM?
>
> The definition of emergent abilities in the multimodal domain is not entirely clear. However, in our recent evaluations, we have indeed observed that our model shows significantly greater improvements in challenging tasks compared to other models, beyond what we see in VQAv2. For example, in the OCR&Spat task defined in MM-Vet, our model scored 59 compared to 14 for InstructBLIP, and in the Rec&Know task, our score was 39 while MiniGPT-4 scored 0.

---

> ### Author Response · Authors · 2023-11-22
> **Kindly Reminder**
>
> Dear Reviewer: Thanks again for your careful and valuable comments! Since the rebuttal discussion is due soon, we’ll be appreciated to know whether our replies have addressed your questions. If there are any further clarifications required or any other concerns, please feel free to contact us. Many thanks!

---

### Official Review · Reviewer_e1q8 · 2023-11-01

**Soundness:** 2 fair
**Presentation:** 3 good
**Contribution:** 2 fair
**Rating:** 6
**Confidence:** 4

**Summary:**

This paper presents CogVLM, a new state-of-the-art vision-language model. To address some issues of previous shallow alignment methods, the authors propose to insert visual expert modules in pretrained language model. Extensive experiments are conducted to evaluate CogVLM, and several SOTA results are achieved.

**Strengths:**

- Strong performance. As tables in the submission, CogVLM shows strong performance compared to other models of equal magnitude. It also exhibits competitive results compared to PaLI-X which has much more parameters.
- Open Source. Open source multimodal fundation model with strong performance has significant impact on the whole society.
- Extensive experiments from different angles and extensive ablation studies. The authors evaluate the superiority of CogVLM in various different kinds of benchmarks (e.g., caption, VQA, text-oriented VQA, grounding, instruction following and etc.).

**Weaknesses:**

- The perhaps biggest weakness with this paper is the writing.
  - This paper starts by raising two possible drawbacks of shallow alignment methods: (i) converge fast but perform worse. (ii) weak visual understanding ability, expecially hallucination. However, both these two disadvantages proposed by the authors are just **hypothesises**, not **compelling** nor **conclusive**. First, the performance gap between BLIP-2 and PaLI-X cames from several possible differences between two framework (e.g., the visual encoder size, the way that visual encoder is pre-trained by). And both MiniGPT-4 and LLAVA have extremely little trainable parameters in the alignment between visual features and language features.
  - Some blanket statements are used. For instance, the author claims that NLP ability is weakened when jointly train the language model in image-text training. However, there are some evidences show that jointly training can benefit both vision task as well as language task, at least in some aspect (e.g., [1]).
  - The motivations and starting points are inconsistent with the experiments. In other words, despite the strong performance, the ablation studies cannot demonstrate that two problems of shallow alignment raised by the writers are well resolved. The ablation studies in Table 6 can prove the effectiveness of CogVLM design. But these numbers cannot prove that deep alignment is better than and solves the issues of shallow alignment, due to the results of shallow alignment method with larger visual encoder (same parameters as vision encoder + vision adapter) are remain unknown.
- Section 2.2 mentions that CogVLM is trained via two-stage process, with 120K and 60K steps respectively. The ablation studies in Table 6 are trained for just 6K steps. However, despite with much fewer iterations, the performance gap between ablation model and the final model is not that significant (e.g., in Table 6, CogVLM achieves 142.8 COCO CIDEr, only ~4 CIDEr score less that the results in Table 3). So does this phenomenone implies that too much iterations in the two-stage training process are unnecessary?
- The visual expert in CogVLM includes FFNs in both attention block and FFN block. Which one is more important for better performance?

[1] Tu, Haoqin, et al. "Sight Beyond Text: Multi-Modal Training Enhances LLMs in Truthfulness and Ethics." arXiv preprint arXiv:2309.07120 (2023).

**Questions:**

- In section 2.3, the author claims that errors in LLAVA-Instruct dataset are corrected by mannual inspection and annotation. Will the corrected dataset be made publicly available?

**Details Of Ethics Concerns:**

N/A.

---

> ### Author Response · Authors · 2023-11-17
> **Response**
>
> Thank you for your comprehensive review and insightful comments. Please allow us to address your points in detail:
>
> > 1. However, both these two disadvantages proposed by the authors are just hypothesises, not compelling nor conclusive. First, the performance gap between BLIP-2 and PaLI-X cames from several possible differences between two framework (e.g., the visual encoder size, the way that visual encoder is pre-trained by). And both MiniGPT-4 and LLAVA have extremely little trainable parameters in the alignment between visual features and language features.
>
> This point is about the effectiveness of deep fusion. The comparison between PaLI-X and BLIP-2 are  just two examples of shallow / deep alignment methods, and not a strict claim. You are right that they cannot be compared directly. We will modify it in the final version to avoid confusion.
>
> > 2. Some blanket statements are used. For instance, the author claims that NLP ability is weakened when jointly train the language model in image-text training. However, there are some evidences show that jointly training can benefit both vision task as well as language task, at least in some aspect (e.g., [1]).
>
> It is not a blanket statement. In our paper, the next sentence of this statement is "According to PaLM-E [2]...", which runs experiments to prove the catastrophic forgetting in LLMs during jointly training. Specifically, Palm-E without freezing the language model parameters during multimodal training, showed a 15% drop in NLU capability and an 87% drop in NLG capability across 21 NLP tasks.  Similar findings were observed in Flamingo, where not freezing the LLM resulted in an average 8% performance decrease across five multimodal benchmarks. Regarding the 'Sight Beyond Text' paper, it only trained with a smaller learning rate on approximately 600,000 data points, which likely led to minimal changes in the LLM parameters, potentially resulting in a significant gap from large-scale pretraining. Hence, it may not be a proper reference to prove no catastrophic forgetting in LLMs during jointly training. For hallucination problems, our model has a significantly higher F1 score on the challenging POPE dataset than other models, which confirms the statement in our paper. See "Results on hard benchmarks" for details.
>
> [2] Danny Driess, Fei Xia, Mehdi SM Sajjadi, Corey Lynch, Aakanksha Chowdhery, Brian Ichter, Ayzaan Wahid, Jonathan Tompson, Quan Vuong, Tianhe Yu, et al. Palm-e: An embodied multi-modal language model. arXiv preprint arXiv:2303.03378, 2023.
> > 3. Ablation studies cannot prove that deep alignment is better than and solves the issues of shallow alignment, due to the results of shallow alignment method with larger visual encoder (same parameters as vision encoder + vision adapter) are remain unknown.
> This proposed ablation study is very reasonable but unfortunately is very hard to implement, because the ViT (CLIP) is pretrained and the performance is not solely determined by the number of parameters. We cannot get a good ViT with arbitrary number of parameters easily for comparison.
>
> However, we can still use a hypothetical comparison. In our paper, the  "VE-full every 4th layer" setting (1.7B) performs much better that the "MLP-adapter"(shallow) setting (140M), (COCO CIDEr 131.2 vs 138.7). If there were a well-trained EVA-CLIP ViT of (5B+1.7B) parameters, it is not possible to increase that much in "MLP-adapter" setting, since the current "MLP-adapter" has already 5B parameters.
>
> If you make the ViT trainable to increase the trainable parameters, the performance will not be better either, which is well-documented in previous works like flamingo[3] (Table 7 (xii) 68.4 ->64.5 after making vision encoder trainable).
> I hope this hypothetical comparsion can ease your concern.
>
> [3] Alayrac, Jean-Baptiste, et al. "Flamingo: a visual language model for few-shot learning." Advances in Neural Information Processing Systems 35 (2022): 23716-23736.
>
> > 4. despite with much fewer iterations, the performance gap between ablation model and the final model is not that significant (e.g., in Table 6, CogVLM achieves 142.8 COCO CIDEr, only ~4 CIDEr score less that the results in Table 3). So does this phenomenone implies that too much iterations in the two-stage training process are unnecessary?
>
> More training iterations are important. Regarding dataset evaluations, the COCO dataset is relatively simple and ground truth is relatively fixed in style and mostly includes common everyday scenes.  Thus, continued training does not show significant improvement, indicating a need for more challenging benchmarks in the research community, like MM-Vet. We observed that our model's performance continued to grow with training, ultimately far surpassing other models. CogVLM achieved a new sota of 52.8 for MM-Vet, which will be updated in our paper. Please see our comment titled "Result for hard datasets" above.

---

> ### Author Response · Authors · 2023-11-17
> **Response2**
>
> > 5. The visual expert in CogVLM includes FFNs in both attention block and FFN block. Which one is more important for better performance?
>
> The ablation experiments in Table 6 suggest that the FFN plays a slightly more significant role than the attention layer. Adding a visual expert every four layers also yielded competitive results. Therefore, choosing a model structure will depend more on the balance between desired model performance and training efficiency.
> > 6.  Will the corrected dataset be made publicly available?
>
> Yes, we will release the dataset if you think this behavior can benefit the community.

---

> ### Author Response · Authors · 2023-11-22
> **Kindly Reminder**
>
> Dear Reviewer: Thanks again for your careful and valuable comments! Since the rebuttal discussion is due soon, we’ll be appreciated to know whether our replies have addressed your questions. If there are any further clarifications required or any other concerns, please feel free to contact us. Many thanks!

---

### Official Review · Reviewer_Yif3 · 2023-11-03

**Soundness:** 3 good
**Presentation:** 3 good
**Contribution:** 2 fair
**Rating:** 5
**Confidence:** 4

**Summary:**

In this paper, the authors studied multimodal LLM and pushed the limit of multimodal LLM by developing a new module called Visual Expert for LLMs. Along with the new model design, the authors curated a large-scale pretraining and instruction-tuning data for the model training. When evaluated on a wide range of vision-language tasks, the proposed model CogVLM exhibits outstanding performance across the board, and surpass models with even much larger size.

**Strengths:**

1. The authors argued that most of the previous multimodal LLMs used shallow connections between vision and models, and thus proposed a new module called visual expert. This new module prompts a more intimate interaction between visual and language tokens in LLMs.

2. The authors curated a large-scale dataset for first-stage pretraining and second-stage instruction tuning. Based on the large-scale training data and the proposed visual expert module, the proposed method achieves a number of state-of-the-art results across a wide range of vision-language tasks.

3. Finally, a number of ablation studies are performed and demonstrate the effectiveness of the proposed method to some extent.

**Weaknesses:**

The main concern to me about this paper is its limited novelty and scientific merit. First of all, the dense interaction between vision and language tokens has been heavily studied prior to the so-called multimodal LLM era. For example, a lot of BERT-style models exploit dense interactions. Second, it is really hard to capture which part is really making the main contribution to the final performance. There are many confounding factors such as the number and type of pretraining data, the instruction-tuning data, different architecture designs, and finetuning strategies. According to Table 6, I can hardly see a clear improvement brought by the introduced new VE modules. The authors start with some good motivation for building more intimate interaction between vision and language, but it finally becomes the emphasis of the benefit of scaling up.

Another missed piece of this work is what we can learn from this work. The state-of-the-art performance should be appreciated. But from the paper, I can hardly tell what the researchers should proceed to further improve the performance. Do we need better model design, or more data and computations? As mentioned in the paper, the authors also used some in-house data, which I guess cannot be released to the public. Given the barrier of reproducing the reported results and also the limited insights delivered by this work, I am sharing a huge concern regarding the current trend of building multimodal LLMs manifested by this work or other related ones.

**Questions:**

As I mentioned above, I have some concerns regarding the scientific merit of this work. I appreciate the effort of pushing the limit of open-sourced multimodal LLMs but do see some potential issues with the current trend of scaling up multimodal LLMs. Given the current state of this paper, I think it is a very good engineering work, but may not be suitable to this research venue.

---

> ### Author Response · Authors · 2023-11-17
> **Response**
>
> Thank you for your comprehensive review and insightful comments. Please allow us to address your points in detail:
> > 1. The main concern to me about this paper is its limited novelty and scientific merit. First of all, the dense interaction between vision and language tokens has been heavily studied prior to the so-called multimodal LLM era.
>
> Our response to this concern is in  1. of "response to common concerns" above. The dense interaction for vision and language is definitely not a new idea, but in the current LLM+ paradigm, it is very valuable to rediscover its effectiveness as a way to better scaling up.
>
> Furthermore, our research extends beyond the mere application of this idea. Through comprehensive ablation studies, we have rigorously examined the influence of various model structures and training strategies on the final outcomes. This includes an in-depth analysis of often-neglected components such as the design of RoPE for image and text sequences, the construction of attention masks, the equilibrium between trainable parameters and model efficiency, and the effect of image self-supervised loss.
>
> It's important to emphasize that each experiment in the realm of large model training is resource-intensive. Therefore, our contribution in identifying and refining effective training methodologies holds significant value in the field.
>
> > 2. I can hardly tell what the researchers should proceed to further improve the performance. Do we need better model design, or more data and computations?
>
> Thank you for your advice. We will add a discussion section in the final version if accepted.
>
> Yes, scaling up is definitely one of the most important ways to increase performance in our opinion. As shown in the Ablation study part, scaling up the model with visual expert is very effective, and the fewer the parameters in the visual expert, the worse the performance.  As you say, how to ensure the **expected performance** after scaling up may be related to model design (like visual expert) or data etc., which is our aim for investigation.
>
> > 3. the authors also used some in-house data, which I guess cannot be released to the public. Given the barrier of reproducing the reported results and also the limited insights delivered by this work, I am sharing a huge concern regarding the current trend of building multimodal LLMs manifested by this work or other related ones.
>
> This is a misunderstanding. All the results except on touchstone are only trained on public datasets, and reproducible. We answer this in  2. of "response to common concerns" above.

---

> ### Author Response · Authors · 2023-11-22
> **Kindly Reminder**
>
> Dear Reviewer: Thanks again for your careful and valuable comments! Since the rebuttal discussion is due soon, we’ll be appreciated to know whether our replies have addressed your questions. If there are any further clarifications required or any other concerns, please feel free to contact us. Many thanks!

---

### Author Response · Authors · 2023-11-17
**Response to Common Concerns**

Dear Reviewers and AC,

Thank you for your valuable feedback and insightful comments,  we would like to address two common concerns here:

> 1. This is a good engineering work, but the research contribution is limited because MoE is not new in multi-modality community and "the performance is mainly attributed to scaling".

 **Response**:
Scaling is not just an engineering work, and not trivial, because a larger model is not necessarily better.

Some methods (e.g. Shallow alignment) will not unleash all the power of scaling up and that's why CogVLM perform better than VLM based on a larger language model such as LLaVA-1.5 and Palm-E-84B. Our *Visual Expert* method, as a successful way to scale up VLM on a pretrained LLM, is of great research value in our opinion.

Many existing applications have indeed explored dense interactions between vision and language tokens in BERT-style models. However, it's important to note that these models typically consist of parameters ranging from 100M to 500M, with the largest, BEiT-3, at 2B. Their training tasks, such as MLM, ITM, and ITC, as well as their model designs, which involve different classification heads for specific tasks, primarily focus on fine-tuning for specialized tasks and understanding intricate image details, such as counting objects in an image. Unlike our approach, these models do not necessitate extensive prior knowledge from strong language models, as BEiT-3 was trained entirely from scratch.

To train a robust VLM based on a pretrained LLM, scaling up with new vision-related parameters while maintaining the language ability is a very different scenario with that in BEiT-3 and similar works. Our method addresses this unique challenge, making it a valuable contribution to the field of vision and language modeling.

We will add a discussion about the differences in the final version.
> 2. "CogVLM is trained with in-house data" and is not reproducible or not fair to compare with other models.

**Response**:
This is a misunderstanding. All the benchmark results except TouchStone are based on the model **without any private data**, i.e. base model, which is clearly listed in our paper (section 3, the first sentence "To rigorously validate the superior performance and robust generalization of our *base model*").

In the first stage of pretraining process, we used the publicly available Laion-5B and COYO-700M datasets. For the second stage of pretraining, the Laion-115M dataset(a subset of Laion) was the primary source. We employed the GLIP-v2 model, also publicly available, to annotate bounding boxes for each image in this dataset. We then finetuned our model on the training set of the benchmarking datasets, which is the standard process to evaluate on these datasets. This approach ensures that our research is easily replicable, promoting transparency and accessibility in the field.
The only private dataset, comprising 11,609 entries, was specifically incorporated to improve the chat version's capabilities in multi-round dialogues. This dataset represents a mere 3% of all publicly available SFT data like LLaVA and LRV-instruction.

We will also adopt the suggestion of reviewer e1q8 to release the manually corrected LLAVA-instruct if accepted.

---

### Author Response · Authors · 2023-11-17
**Results on hard benchmarks**

To further show the emergent ability in our model, we report the zero shot result on 3 hard datasets.

To show that we can achieve good performance on both benchmarks and real-world chat, the evaluated checkpoint are trained on the mixture of VQA datasets and SFT data. (Response to the Reviewer c9aL Question 2)

| Method                   | LLM          | MM-VET | POPE(Adv)  |  TouchStone |
|--------------------------|--------------|--------|-------|-------|
| BLIP-2 | Vicuna-13B   | 22.4   | -     | - |
| Otter | MPT-7B       | 24.7   | -     | - |
| MiniGPT4 | Vicuna-13B   | 24.4   | 70.4  | 531.7 |
| InstructBLIP | Vicuna-13B   | 25.6   | 77.3  | 552.4 |
| LLaVA | LLaMA2-7B   | 28.1   | 66.3  | 602.7 |
| LLaMA-Adapter v2 | LLaMA-7B     | 31.4   | -     | 590.1 |
| mPLUG-Owl | LLaMA-7B | -| 66.8| 605.4 |
| DreamLLM | Vicuna-7B     | 35.9   | 76.5  | - |
| LLaVA-1.5 | Vicuna-13B   | 36.3   | 84.5  | - |
| Emu | LLaMA-13B     | 36.3   | -     | - |
| Qwen-VL-Chat    |  Qwen-7B |  - |  - | 645.2 |
| CogVLM               | Vicuna-7B    | **52.8**   | **87.6**  | **742.0** |

---

> ### Public Comment · ~Nick_Yang1 · 2023-11-27
>
> This study leaves me somewhat unconvinced that the improvements are primarily due to the architectural alterations. I have three primary concerns regarding this:
>
> 1. The unavailability of training data raises questions. The approach takes advantage of substantial proprietary training data during pre-training, which introduces the possibility of test data leakage during this phase.
>
> 2. The use of an oversized visual encoder (4.8B) seems unfair in the comparison. It would be more insightful to use standard vision transformers like ViT-L, g, or G for a more balanced comparison. This calls into question whether the improvements are largely due to the colossal visual backbone.
>
> 3. The unavailability of the training code raises doubts about potential tweaks during the training process. It would be more reassuring if the full training procedure could be reproduced, eliminating any uncertainties about the process.
>
> Therefore, unless these problems are solved, I think it is not unconvincing for publication, personally.

---

> ### Public Comment · ~Lilong_Xue1 · 2023-11-28
> **Public Comment to Nick Yang's comment**
>
> 1. Firstly, the training data used for model pre-training is transparent in their paper, such as LAION and COYO-700M datasets. The majority of current multimodal models have utilized these datasets, either during the training of the visual encoder or in the image-text alignment phase. Your concerns about data usage are unfounded and somewhat offensive. Secondly, it's plausible that the research team implemented strict separation between training and testing data, ensuring that the data used during the testing phase is completely independent of the training set. You only need to use the model's demo to find that its effect is indeed very strong.
> 2. I think the use of a 4.8B parameter visual encoder was to explore and demonstrate the potential of model performance at a high parameter scale. This doesn't imply that smaller models couldn't achieve similar improvements. In fact, the authors have conducted a comparative analysis with an OpenCLIP-Large model, as highlighted in their response to reviewer c9aL. The outcomes of this comparison reveal only a marginal difference in performance (84.7 vs 83.4 on VQAv2), which further underscores the point that substantial advancements are not solely contingent on the size of the model.
> 3. The paper in question likely provides ample detail, enabling interested researchers to effectively replicate the model. Additionally, the authors have made significant efforts towards transparency and accessibility by releasing the training code and model checkpoints. For those interested in delving deeper into the specifics of their work. This proactive approach by the authors greatly facilitates further research and exploration in the field.
>
> Therefore, since these problems are solved, I think it is convincing for publication, personally.

---

> > ### Public Comment · ~Nick_Yang1 · 2023-11-29
> >
> > With reference to LLaVA-1.5, it offers a straightforward reproduction of the results via pre-training and instruction tuning code. However, for CogVLM, we cannot guarantee the reproducibility of the results unless the complete training code and logs are disclosed. The training code released on Github does not correspond to any experiments illustrated in the paper. It merely provides a basic example of code execution rather than a demonstration of the pre-training process itself.
> >
> > Furthermore, the increased size of the model doesn't seem promising, given that the number of model parameters has doubled. This scaling is not conducive to further size escalation. For instance, imagine having an LLM with 200B parameters. To achieve this, you would need to train a model with over 400B parameters, which lacks elegance.
> >
> > In addition, the author only presents the fine-tuning results on VQA concerning the marginal improvement of the vision encoder. It's well-known that fine-tuning involves various tricks and doesn't fully demonstrate the model's capability. Why not display the model's zero-shot capabilities on benchmarks like MME, MMBench, or MM-Vet? This would allow us to determine whether the primary improvement stems from the larger backbone. I believe this aspect significantly influences performance, especially in a zero-shot context.

---

> > > ### Public Comment · ~Lilong_Xue1 · 2023-11-29
> > > **Public Comment to Nick Yang's comment**
> > >
> > > 1. The authors have already released a codebase sufficient to reproduce their model. Anyone can easily construct a dataset and utilize the code from Github for training and inference to get results, so your ambiguity on 'reproduction' isn't clear.
> > > 2. The authors consistently emphasize the importance of balancing computational efficiency and model effectiveness, substantiated by thorough experimentation in the ablation section of the paper. There's no declared rule that an additional 200B parameters are necessary for a 200B language model. By adding extra parameters every eight layers, akin to how Flamingo operates, this model would require only 25B parameters. If one were to apply methods similar to LoRA or employ a smaller hidden size, the training parameters could further diminish to just a few billions, rendering it fully acceptable.
> > > 3. 'It's well-known that fine-tuning involves various tricks and doesn't fully demonstrate the model's capability.' I'm afraid this perception is not as 'well-known' as suggested. Essential research papers in the deep learning realm have proven their effectiveness through fine-tuned results, such as ResNet, BERT. Are these merely tricks? The real well-known fact is that for datasets like VQAV2, OKVQA, the prediction results must match precisely, implying that the language style of the output prediction significantly impacts the outcome. Hence, appropriately fine-tuning the model to adapt to a dataset's language style can indeed better reflect the model's capabilities. To further support this, the EVA CLIP paper did conduct some zero-shot experiments: the 0.3B-parameter CLIP-Large model scored 79.8 on ImageNet in a zero-shot context, while the 4.4B-parameter CLIP-E model scored 82.0. This closely resembles the authors' fine-tuned results on the VQAV2 and Visual7W datasets. On the contrary, your belief that 'this aspect significantly influences performance, especially in a zero-shot context.' lacks empirical grounding. Considering your biased critique, it raises suspicion that you may have a vested interest in some of the models compared within the paper, leading to your persistent nitpicking.

---

### Meta-Review · Area_Chair_MTwn · 2023-12-14

**Metareview:**

This paper was reviewed by three experts and received mixed scores. Though all reviewers agree some aspects of the paper are promising and the rebuttal has addressed some questions, they also consistently raise concerns listed below.

1. The technical contribution of the proposed model is incremental (Yif3, c9aL).

2. The motivation and insights are unclear (Yif3, e1q8).

3. The clarity of presentations needs to be improved ( Yif3, e1q8 ).


While the research demonstrated indeed has promise, the decision is not to recommend acceptance in its current state. The authors are encouraged to consider the reviewers' comments when revising the paper for submission elsewhere.

**Justification For Why Not Higher Score:**

1. The technical contribution of the proposed model is incremental (Yif3, c9aL).

2. The motivation and insights are unclear (Yif3, e1q8).

3. The clarity of presentations needs to be improved ( Yif3, e1q8 ).

**Justification For Why Not Lower Score:**

NA

---

### Decision · Program_Chairs · 2024-01-16

Reject